

# The self-consistent quantum-electrostatic problem in strongly non-linear regime

Pacome Armagnat, A. Lacerda-Santos, Benoit Rossignol,
Christoph Groth and Xavier Waintal⋆

Univ. Grenoble Alpes, CEA, IRIG-PHELIQS GT, F-38000 Grenoble, France

⋆ xavier.waintal@cea.fr

## Abstract

The self-consistent quantum-electrostatic (also known as Poisson-Schrödinger) problem is notoriously difficult in situations where the density of states varies rapidly with energy. At low temperatures, these fluctuations make the problem highly non-linear which renders iterative schemes deeply unstable. We present a stable algorithm that provides a solution to this problem with controlled accuracy. The technique is intrinsically convergent even in highly non-linear regimes. We illustrate our approach with both a calculation of the compressible and incompressible stripes in the integer quantum Hall regime as well as a calculation of the differential conductance of a quantum point contact geometry. Our technique provides a viable route for the predictive modeling of the transport properties of quantum nanoelectronics devices.


# 1 Introduction: accurate modeling of quantum nanoelectronics

The control of quantum-mechanical systems in condensed matter has reached a level of maturity where researchers seek to further develop these systems into full-fledged quantum technologies that provide the building blocks for complex devices. As part of this endeavor it is necessary to develop simulation tools that allow to predict the properties of such devices. This stage has been already reached for some quantum technologies. For example, the theoretical description of devices based on superconducting circuits has become reliable enough to be used for their conception [1]. In contrast, an accurate predictive modeling of semi-conductor-based circuits turns out to be much more challenging [2–4].

The difficulty with semi-conductors is that the presence of a strong electric field effect (which is precisely what makes them so useful) is associated with the presence of two vastly different energy scales. On one hand the various band offsets lie in the 1-eV-range (this is also the typical voltage applied on the electrostatic gates to deplete an electron gas). On the other hand, the typical Fermi energy of a two-dimensional electron gas in an heterostructure (2DEG) lies in the 1-meV-range, i.e. a scale almost three orders of magnitude smaller than the one above. Addressing this multi-scale quantum-electrostatic problem [5] is not an easy task, yet it is a prerequisite for the development of quantum technologies such as quantum-dot-based localized qubits [6] or flying qubits [7].

This article presents a new technique for solving the quantum-electrostatic problem that is both accurate (i.e. it is able to deal with realistic energy scales in the 10-100 μeV range) and robust (i.e. its convergence must not rely on the fine tuning of the parameters of the algorithm). Furthermore, our technique is general-purpose, i.e. it applies to a wide spectrum of materials (semi-conducting heterostructures but also nanowires, graphene like materials, topological materials) and geometries (hybrid systems, multi-terminal devices).

In its simplest mean field form, the quantum-electrostatic problem can be formulated as the solution of a self-consistent set of three equations. For a given electronic density, the solution of the Poisson equation (i) provides the electrostatic potential. For a given electrostatic potential, the solution of the Schrödinger equation (ii) provides the energy spectrum and wave functions. Statistical physics provides the last equation: filling up the spectrum according to the Fermi distribution (iii), yields the electronic density. This problem, hereafter referred as the (self-consistent) quantum-electrostatic problem, has a long history in both physics and chemistry: it lies at the heart of both material science and quantum chemistry and is in particular the central problem solved in density-functional theory calculations [8]. The problem is also known as the self-consistent Hartree approximation as well as the self-consistent Poisson-Schrödinger equation. It can be seen as the first step of a systematic treatment of the many-body effects associated with Coulomb repulsion. The vast majority, if not all, of the approaches

to its solution use some form of iterative scheme: one calculates the potential from the density (electrostatic problem), then the density from the potential (quantum problem) and so on until convergence. Earlier approaches used straightforward iterations [9–11]. However, faster convergence can be obtained by combining several previous approximate solutions to form a new one in some form of mixing. Mixing approaches include simple under-relaxation [12,13], Direct Inversion in the Iterative Subspace (DIIS) [14], Anderson [15] or Broyden mixing [16]. Better converging properties can be obtained using root finding methods, such a variations on the Newton-Raphson algorithm which can be implemented either with an exact Jacobian [17] or an approximate one [18,19]. The most sophisticated approaches use different predictor-corrector algorithms where an approximate problem (often within the Thomas-Fermi approximation) is solved to obtain predictions of the solution which are corrected iteratively by solving the full equations [20–26].

Although these approaches have been successful in various contexts, in particular when the temperature is not too low [27] or when the density of states is rather smooth, they also fail spectacularly even in simple situations where the density of states has rapid variations in energy such as in the quantum Hall regime. When they do work, they often necessitate manual fine tuning of parameters in order to converge, or even require deep physical insight to come up with a good approximation of the result that can be used to attain convergence. In contrast, the method presented in this article is stable in these highly non-linear situations.

An important distinction must be made between gapped systems such as band insulators or molecules and conducting systems such as metals or semi-conductors [8]. In the former, the filling of the quantum states is unambiguous; the absence of available states at the Fermi level makes screening impossible. Solving the quantum-electrostatic problem for these systems is relatively easy since most iterative algorithms converge. The second situation, the quantum-electrostatic problem for conductors, combines the double difficulty of being non-local (long range Coulomb repulsion) and non-linear (the electronic density depends on the square of the wave-function). It is the focus of this article.

Our approach takes a fresh perspective on the problem: instead of looking for self-consistency iteratively, we obtain self-consistency exactly for an approximate problem. This approximate problem is already very close to the exact one and can be brought arbitrarily close iteratively. The main advantage of this point of view is that the self-consistent approximate problem can be solved to arbitrary precision at no significant computational cost; its solution is provably intrinsically convergent.

We start this article by formulating the self-consistent quantum-electrostatic problem in Sec. 2. In Sec. 3, we address a simple yet illuminating zero-dimensional model that may be solved exactly. In Sec. 4 we formulate the adiabatic self-consistent problem that forms the backbone of our method. How to use the adiabatic problem to solve the initial self-consistent quantum-electrostatic problem is explained in Sec. 5. Our algorithm requires solving a generalization of the standard electrostatic problem which is explained in Sec. 6. Sec. 7 deals with the last technical difficulty, the numerical integration of the local density of states. The last two sections are devoted to two applications of our method. The first is the study of the compressible/incompressible stripes in the quantum Hall effect (Sec. 9) and the second is the calculation of the conductance in a quantum point contact geometry (Sec. 10).

## 2 Formulation of the self-consistent quantum-electrostatic problem

Let us formulate the quantum problem. We consider a non-interacting Hamiltonian $H$ that describes a quantum conductor. It can consist of a scattering region connected to electrodes as

in typical quantum transport problems [28], it can also describe bulk physics in one (infinite nanowires), two (two-dimensional electron gas, graphene) or three dimensions. All these systems share an important property. They are infinite, hence possess a proper density of states as opposed to a discrete spectrum. We suppose that $H$ has been discretized onto sites $i$ filled with the electronic gas. This discretization can be obtained in various ways. One can discretize an effective mass or $k \cdot p$ Hamiltonian; one can also construct a tight-binding model by projecting a microscopic Hamiltonian onto atomic orbitals. The electron gas is subject to an electrostatic potential $U(\vec{r})$ whose discretized form is written as a vector $U$ of components $U_i$. The Schrödinger equation reads

$$\sum_{j \in \mathcal{Q}} H_{ij}\psi_{\alpha E}(i) + U_i \psi_{\alpha E}(i) = E\psi_{\alpha E}(i), \tag{1}$$

where $\psi_{\alpha E}(i)$ is the electronic wave-function at energy $E$ and the discrete index $\alpha$ labels the different bands (or propagating channels) of the problem. In the actual simulations performed in this paper, $\psi_{\alpha E}(i)$ have been calculated with the Kwant package [28]. We call $\mathcal{Q}$ the set containing all the sites on which the quantum problem is defined. The number of electrons on site $i \in \mathcal{Q}$ is given by filling up the states with the Fermi distribution $f(E) = 1/[e^{E/(k_B T)} + 1]$ (hereafter the Fermi energy $E_F = 0$ is our reference energy),

$$n_i = \int dE \rho_i(E)f(E), \tag{2}$$

where we have introduced the local density of states (LDOS),

$$\rho_i(E) \equiv \frac{1}{2\pi}\sum_{\alpha}|\psi_{\alpha E}(i)|^2. \tag{3}$$

The last equation that closes the problem is the Poisson equation that reads

$$\nabla \cdot (\epsilon(\vec{r})\nabla U(\vec{r})) = \frac{-e}{\epsilon}[n(\vec{r}) + n^{\mathrm{d}}(\vec{r})], \tag{4}$$

where $e$ is the electron charge, $\epsilon$ the local dielectric constant and $n(\vec{r})$ is the density of the electron gas. The $n^{\mathrm{d}}(\vec{r})$ term corresponds to any charge density located elsewhere in the system, e.g. dopants or charges trapped in an oxide. The Poisson equation is also specified by its boundary conditions. We shall use Neumann conditions at the boundary of the system as well as Dirichlet conditions at the electrostatic (metallic) gates. As for the quantum problem, we suppose that the Poisson equation has been discretized with some scheme such as a finite difference, finite element or (as we have done, see Sec. 6) finite volume method. The discretization of the Poisson equation is rather straightforward and most approaches converge smoothly to the correct solution. The discretized Poisson equation takes the form

$$\sum_{\nu \in \mathcal{P}} \Delta_{\mu\nu}U_\nu = n_\mu + n_\mu^{\mathrm{d}}. \tag{5}$$

We call $\mathcal{P}$ the set containing all sites of the system on which the Poisson equation is defined. We emphasize that the quantum problem is defined on a subset of the electrostatic problem, i.e. $\mathcal{Q} \subset \mathcal{P}$. The set $\mathcal{P} \setminus \mathcal{Q}$ contains regions with dielectric materials, dopants or vacuum. We often use greek letters for sites $\mu \in \mathcal{P}$ and latin letter for sites $i \in \mathcal{Q}$.

The problem of (partial) dopant ionization is commonly addressed by supposing that they correspond to a certain number $n_\mu^0$ of localized levels with degeneracy $g$ and energy $E_0$ so that

$$n_\mu^{\mathrm{d}} = \frac{n_\mu^0}{1 + g e^{\frac{U_\mu + E_0}{k_B T}}}. \tag{6}$$

At very low temperature, the focus of this paper, this equation can only have three solutions: the dopants are fully ionized $n_\mu^d = n_\mu^0$; no dopants are ionized $n_\mu^d = 0$; or $U_\mu = -E_0$. In the first two regimes Eq. (6) fixes the charge density in the Poisson equation. In the last one the dopant layer acts as an effective electrostatic gate, i.e. as a Dirichlet boundary condition in the Poisson equation. For the problems studied here, we restrict ourselves to the experimentally relevant regime where the dopants are fully ionized.

The set of equations (1), (2), (3) and (5) forms the (discrete version of the) quantum electrostatic problem. Hereafter, its Full Self-Consistent solution is referred to as FSC .

In what follows, our approach will be illustrated with a two-dimensional electron gas (2DEG) formed at the interface between GaAs and GaAlAs [29]. We model the 2DEG within the effective mass approximation by discretizing

$$\frac{1}{2m^\star}\left(i\hbar\vec{\nabla} - eA\right)^2 \psi + eU(x,y)\psi = E\psi \tag{7}$$

on a regular grid using Peirls substitution. We have used a discretization step of 20 nm for the 3D calculations and 6.3 nm for the 2D calculations. The vector potential $\vec{A}$ is taken in the Landau gauge associated with a perpendicular magnetic field $\vec{B} = \vec{\nabla} \times \vec{A}$. The effective mass $m^\star$ is set to $0.067\,m_e$. Furthermore, we assume the permittivity $\epsilon$ to be $\epsilon = 12\epsilon_0$ in the semi-conductors. The dopant concentration is adjusted to obtain a 2DEG density equal to $n = 2.11 \times 10^{11}$ cm$^{-2}$. We use Neumann boundary conditions at the boundary of the Poisson simulation box and Dirichlet at the electrostatic gates. The two geometries that will be considered are shown in Fig. 1 (b) and (c).

## 3 Role of non-linearities: a zero-dimensional model

Let us start with a very simple zero-dimensional problem that already provides key insights into the structure of the quantum-electrostatic problem. We consider an infinite homogeneous 2DEG characterized by a – spatially invariant – density $n$ and an electric potential $U$. The system is sketched in Fig. 1(a). An electrostatic gate placed at a distance $d$ above the 2DEG forms a planar capacitor with the latter. The Poisson equation for this problem is readily solved: it is given by the solution of the infinite planar capacitor problem:

$$n = \frac{\epsilon}{ed}[V_g - U], \tag{8}$$

where $V_g$ is the electric potential at the electrostatic gate. The quantum problem is also readily solved. At zero temperature, n is given by the integrated density of states (ILDOS):

$$n = \int^\mu dE\rho(E), \tag{9}$$

where $\mu$ is the chemical potential. At equilibrium, the total *electrochemical* potential of the 2DEG has a fixed value $U - \mu/e = 0$ which is our reference potential. The two equations (8) and (9) form the set of equations to be solved self-consistently. At zero magnetic field, the density of states (DOS) is constant $\rho = m^\star/(\pi\hbar^2)$ and these equations reduce to a trivial linear system of equations. The situation is more interesting when one switches a magnetic field $B$ perpendicular to the 2DEG. Indeed, in presence of a magnetic field, the DOS consists of Dirac peaks at the positions of the Landau levels. The system reduces to

$$n = \frac{\epsilon}{ed}[V_g - \mu/e], \tag{10}$$

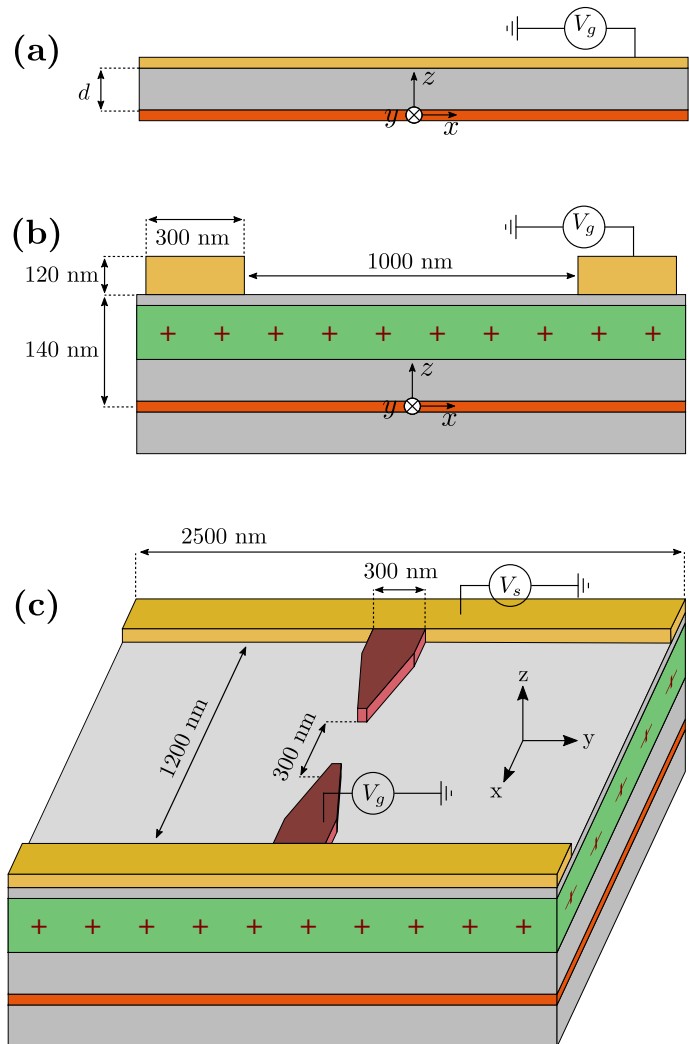

Figure 1: Schematics of the three systems considered in this article. (a) Infinite 2DEG along $x$ and $y$ directions. (b) Quasi-one dimensional wire infinite along the $y$ direction. (c) Quantum Point Contact geometry. The red part corresponds to the 10 nm thick 2DEG. The green part corresponds to the doping region. The yellow and dark red part correspond to the electrostatic gates. The gray part corresponds to effective dielectrics (here GaAs and GaAlAs).

and

$$n(\mu) = \frac{2eB}{h} \sum_{n=0}^{\infty} \theta(\mu - E_n), \tag{11}$$

with $\theta$ the Heaviside function, $E_n = \hbar\omega_c\left(n + \frac{1}{2}\right)$ the energies of the Landau levels and $\omega_c = eB/m^\star$ is the cyclotron frequency.

Fig. 2 shows the two functions $n$ versus $\mu$ for the Poisson problem Eq. (10) (blue line) and quantum problem Eq. (11) (orange line). Solving the self-consistent equations amounts to finding the intersection point of these two curves. This is a trivial task where the accuracy of the solution increases exponentially with the number of evaluations of the two functions: one curve (Poisson) is strictly decreasing with $\mu$ while the other (Quantum) is strictly increasing so that a simple dissection scheme converges exponentially.

Using this 0D model, one can also verify that iterative algorithms are extremely unstable in

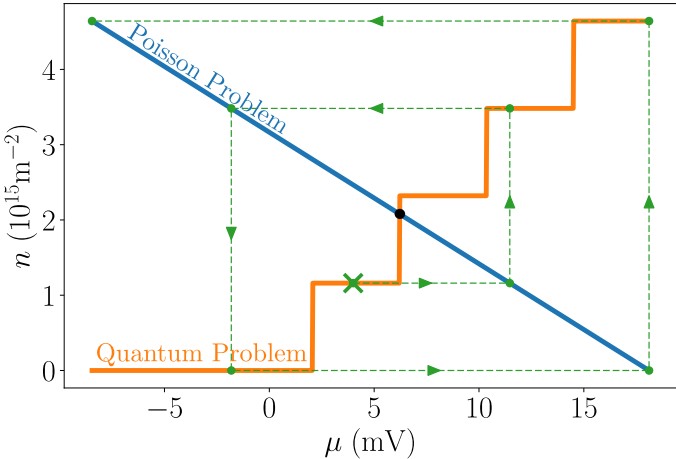

Figure 2: Zero-dimensional model for the self-consistent quantum electrostatic problem in the planar capacitor 0D geometry of Fig. 1(a). Orange line: solution of the quantum problem Eq. (11). Blue line: solution of the Poisson problem Eq. (8). Green arrows: example of a simple iterative solution of the problem which fails to converge. Geometric capacitance $C = 0.028$ F/m$^2$, dopant density $n_0 = 3.16 \times 10^{11}$ cm$^{-2}$, magnetic field $B = 2.4$ T.

presence of magnetic field. For instance, the green arrows indicate a simple iterative scheme where one starts with a given chemical potential, calculates the density from the ILDOS, then gets the potential from Poisson. One applies the preceding sequence iteratively until convergence. The non-linearity of the ILDOS – which reflects the rapid variation of the DOS – makes this scheme divergent even with a good initial guess for the density. This is a rather extreme (yet physical) situation where the ILDOS has a highly non-linear staircase shape. Yet, even under more favorable conditions, the convergence of iterative schemes is seldom guaranteed and one has to rely on the fine-tuning of the parameters of the algorithm to obtain reliable results. These parameters characterize e.g. the learning rate or approximate solutions used by the algorithm to speed up convergence.

In the next two sections, we introduce our algorithm for solving the full (spatially dependent) problem. Conceptually, the idea is to reduce the global self-consistent problem to a set of approximate local self-consistent problems similar to Eq. (8) and Eq. ( 9).

## 4 The adiabatic self-consistent problem

The zero-dimensional model of Sec. 3 could be solved exactly – even in the presence of strong non linearities – because finding its solution amounted to searching for the intersection between two curves. In this section we will introduce the adiabatic self-consistent problem. It is a local problem where on each site $i \in \mathcal{Q}$ one needs to solve an intersection problem similar to the one in Sec. 3. Hence, it can be easily solved numerically.

The adiabatic self-consistent problem is obtained by making two hypotheses. The first concerns the quantum problem and is called the quantum adiabatic approximation (QAA). The second is applied to the Poisson problem and is named the Poisson adiabatic approximation (PAA). The adiabatic self-consistent problem is similar in spirit to the approximate problem solved in density functional theory within the Local Density Approximation (LDA) [30]. The LDA becomes exact in the limit of an infinitely spatially smooth electronic density. Similarly, the adiabatic self-consistent problem becomes exact when the electric potential is infinitely

smooth. However, the error of LDA cannot be controlled. In contrast, we can systematically improve the adiabatic self-consistent problem until its solution matches the FSC solution.

The adiabatic self-consistent problem will be our main tool to solve the self-consistent quantum electrostatic problem defined in Sec. 2. In the current section, we show how to formulate and exactly solve the adiabatic self-consistent problem. In Sec. 5, we will show how to use the adiabatic self-consistent problem to obtain the FSC solution.

## 4.1 Quantum Adiabatic Approximation (QAA)

The quantum adiabatic approximation (QAA) maps the quantum problem onto a local problem. We consider an electric potential $U_i$ defined on the quantum site $i$, with $i \in \mathcal{Q}$. We suppose that we have solved the Schrödinger equation (1) for this potential and computed the LDOS $\rho_i(E)$ on each site $i$ using Eq. (3). The density $n_i$ is obtained by filling up the states according to Eq. (2). Now suppose that we introduce a perturbation $\delta U$. The electric potential becomes $U + \delta U$, i.e. $U_i \rightarrow U_i + \delta U_i$. One should thus recalculate $n_i[\delta U]$. In principle, this would imply re-solving the Schrödringer equation for $U + \delta U$, which is a computationally expensive task. Also, the new value of $n_i[\delta U]$ depends on $\delta U_j$ in a non-local way ($j \neq i$). However, if $\delta U$ is either small or has very smooth spatial variations, one can use the Quantum Adiabatic Approximation (QAA),

$$n_i[\delta U] \approx \int dE \, \rho_i(E) f(E + \delta U_i). \tag{12}$$

In the QAA, one needs not recalculate the LDOS. Eq. (12) is exact to first order in $\delta U$ (small perturbation). It is also exact when $\delta U$ is infinitely smooth (when $\delta U_i$ does not depend on $i$, a global shift in energy does not modify the wave functions). We shall find empirically that the QAA is an excellent approximation for realistic systems. Indeed, effective electrostatic potentials *do* vary smoothly, the rapidly varying part of the electric potential at the atomic level being usually included in a renormalization of the effective parameters of the theory. Note that with our convention the electrochemical potential is set to zero so that a change of electric potential $\delta U_i$ is equivalent to the opposite change in the local chemical potential, i.e. $\delta U_i + \delta \mu_i = 0$. The QAA approximation bears two important features: (i) it is a local equation on each site $i$ and (ii) the knowledge of the LDOS is sufficient to calculate $n_i$ for any variation $\delta U$.

In practice, we shall construct an interpolant of $\rho_i(E)$ in order to calculate the integral Eq. (12) for various $\delta U_i$. At zero temperature, Eq. (12) reduces to the integrated local density of states (ILDOS),

$$n_i[\delta \mu_i] \approx \int^{\delta \mu_i} dE \, \rho_i(E). \tag{13}$$

The shape of the LDOS often contains $1/\sqrt{E}$ singularities (no magnetic field) or Dirac functions $\delta(E)$ (Landau levels in presence of magnetic field). This is illustrated in Fig. 3 where we have plotted the functions LDOS and ILDOS versus energy for two magnitudes of the magnetic field. At low magnetic field the integration can be performed with quadrature techniques. At large magnetic field, however, a different approach is required to handle the presence of the Dirac peaks. This aspect is discussed in Sec. 7.

## 4.2 Poisson Adiabatic Approximation (PAA)

The Poisson adiabatic approximation (PAA) maps the Poisson problem onto a local problem. The exact solution of the Poisson problem can be formally written as

$$U_i = \sum_j G_{ij} n_j + U_i^s, \tag{14}$$

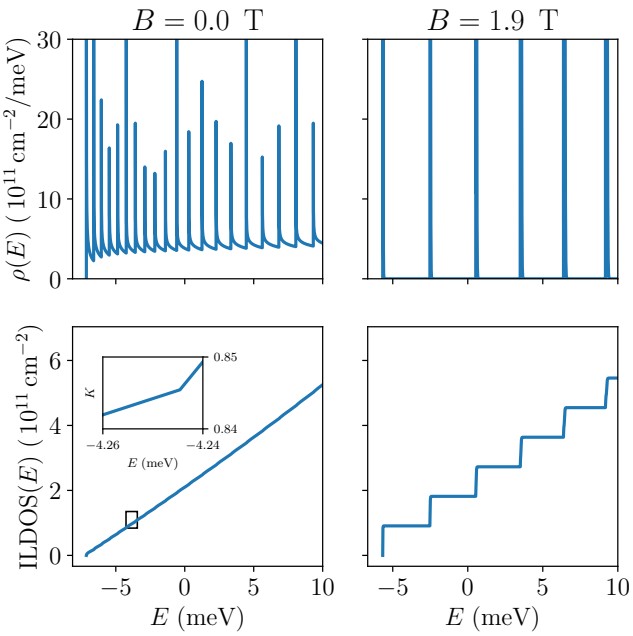

Figure 3: Top: Local density of states $\rho_i(E)$ (LDOS) at the center of the gas ($x = 0$) under $B = 0$ T (left) and $B = 1.86$ T (right) as a function of energy for the geometry of Fig. 1(b). Bottom: Integral of the local density of states (ILDOS) for the same magnetic fields. The gate voltage is $V_G = -1$ V. Inset: zoom of the main curve showing the cusp created by the $1/\sqrt{E}$ singularity of the DOS

with $i, j \in \mathcal{Q}$. The matrix $G$ is (a discretized version) of the Green function of the Poisson equation and $U_i^s$ accounts for the source terms in the problem. It is important to note that Eq. (14) is defined only on the sites $i \in \mathcal{Q}$ where the quantum system lies, i.e. the extra sites $\mu \in \mathcal{P} \setminus \mathcal{Q}$ have been integrated out. In the continuum $G$ is essentially $e^2/(4\pi\epsilon|r - r'|)$, although it may decay faster at long distances due to the screening effect of the electrostatic gates. We invert the matrix $G$ and obtain

$$n_i = \sum_{j \in \mathcal{Q}} C_{ij} U_j + n_i^s, \tag{15}$$

where $C = G^{-1}$ is the capacitance matrix and $n^s = -CU^s$ accounts for the source terms. Eq. (15) has a very similar structure to the Poisson equation (8). However, it is only defined on the site $i \in \mathcal{Q}$. The $C$ matrix is a central object of our approach. How to compute its relevant elements will be explained in Sec. 6.

Fig. 4 shows the elements of the $G$ and $C$ matrices calculated for the geometry in Fig. 1(b). As expected, the Green function $G$ is highly non-local: a change in $n_i$ has an effect on $U_j$ over a large distance. In sharp contrast, the $C$ matrix is extremely local. Indeed, to a good approximation, the $C$ matrix is the discretized version of the Laplacian, hence a local operator. This statement would be mathematically exact if we had not integrated out any sites, i.e. $\mathcal{Q} = \mathcal{P}$. The locality of the capacitance matrix $C$ is the central property on which PAA is based. In the Poisson Adiabatic Approximation (PAA), we assume that the change $\delta U_i$ is smooth so that we can approximate Eq. (15) with

$$n_i[U + \delta U] \approx n_i[U] + C_i \delta U_i, \tag{16}$$

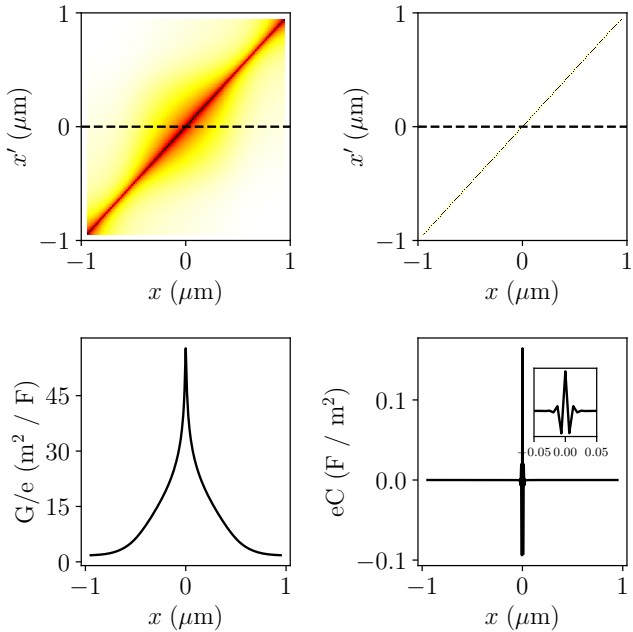

Figure 4: Green's function $G$ (left) and Capacitance matrix $C$ (right) for the geometry shown in Fig. 1(b). Top panels: 2D colormaps of $G_{xx'}$ (left) and $C_{xx'}$ (right). Lower panels: 1D cuts $G_{x,0}$ (left) and $C_{x,0}$ (right). Inset: zoom of the lower right panel. The $C$ matrix is very local while the $G$ matrix is not.

where the local capacitance $C_i$ is defined as

$$C_i = \sum_j C_{ij}. \tag{17}$$

Eq. (16) is exact in the limit where $\delta U_i$ can be considered as constant on the scale of the support of $C$. As we shall see, PAA is generally an excellent approximation, with a small caveat explained in Sec. 5.

The applications studied in this manuscript correspond to 1D quantum problems embedded in 2D electrostatic problems (or 2D embedded in 3D). In such cases, $\sum_j C_{ij} \neq 0$ for all $i \in \mathcal{Q}$ and the above definition of $C_i$ is sufficient. More generally, one could also study a 2D quantum problem embedded in a 2D electrostatic problems (or 3D embedded in 3D). In such systems there will be "bulk" sites where $\sum_j C_{ij} = 0$. A bulk site is defined as a site surrounded only by other $\mathcal{Q}$ sites. That is, $i \in \mathcal{Q}$ is a bulk site if and only if for all $\mu$ such that $\Delta_{\mu i} \neq 0$, one has $\mu \in \mathcal{Q}$. The presence of these bulk sites where $C_i = 0$ can generate convergence problems (in the step II defined in Sec.5). Hence, for bulk sites, it is necessary to enforce a small finite value of $C_i$ by defining

$$C_i = \frac{\Delta_{ii}}{\Gamma}, \tag{18}$$

where $\Gamma$ is a numerical constant. We have found emprically that values of $\Gamma \approx 100$ provide a fast and robust convergence of bulk systems but we defer their study to a subsequent publication.

## 4.3 Solving the local self-consistent problems

Together, Eq. (12) and (16) form a local self-consistent problem on every site $i \in \mathcal{Q}$. This is the adiabatic self-consistent problem. Solving this set of equations simply amounts to finding the

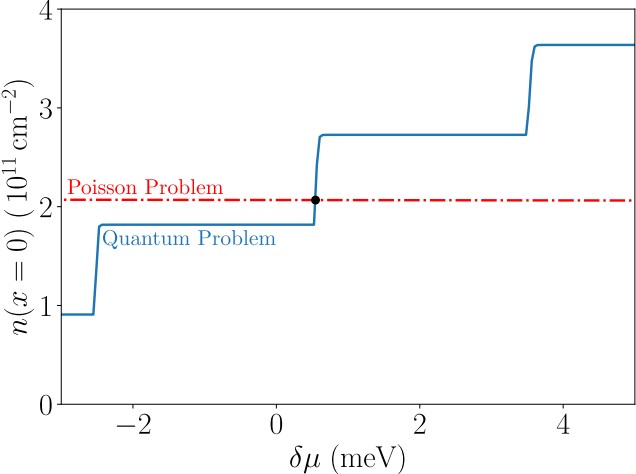

Figure 5: Solution of the local self-consistent problem at $x = 0$, $B = 1.87\,T$ and with $V_G = -1$ V for the geometry of Fig. 1(b). Blue line: ILDOS Eq. (13) versus chemical potential $\delta\mu$. Orange dashed line: local Poisson problem Eq. (16) versus $\delta\mu = -\delta U$. The intersection of the two curves is the solution of the (local) adiabatic self-consistent problem.

intersection of the Poisson and Quantum curves for every site, which can be done extremely efficiently. More importantly, the solution always exist and can always be found with exponential accuracy. In practice, any one-dimensional root finding routine works very efficiently.

Fig. 5 shows an example of the adiabatic self-consistent problem for a given site $i \in \mathcal{Q}$, where we have used the bulk DOS Eq. (11) as the LDOS. This problem and the zero-dimensional model of Sec. 3 are solved in a similar way. The only difference is that in the adiabatic self-consistent problem a different intersection must be found for each site $i \in \mathcal{Q}$. Observe that the electrostatic Eq. (16) is almost an horizontal line, i.e. the density depends only weakly on the potential on this scale. This is a consequence of the electrostatic energy being much larger than the kinetic energy. A direct consequence is that the convergence of the density is achieved very rapidly, before one obtains the converged potential. A secondary consequence is that one should chiefly monitor the convergence of the potential, a more sensitive quantity than the density.

## 5 Relaxing the Adiabatic self-consistent problem

The PAA and QAA approximations have been designed such that the initial global self-consistent problem can be reduced to a set of local problems that can be solved exactly and efficiently. In this section we propose an algorithm to relax these two approximations, and thus obtain the FSC solution of the full quantum-electrostatic problem. The convergence towards the exact solution is achieved by iteratively improving the local problems until they match the global one. Although this relaxation is iterative, one iterates on the adiabatic self-consistent *problem*, in contrast to iterating on the *solution* as is usually done. In practice, we observe extremely fast convergence, typically in a single iteration of the quantum problem (the computational bottleneck calculation). The relaxation of PAA and QAA is done using three relaxation steps, **I**, **II** and **III**, which will be now detailed.

**I.** In Sec. 4 we have argued that the Poisson approximation is generally accurate. There is, nonetheless, a caveat to this argument. In fact, the PAA is of very high accuracy inside

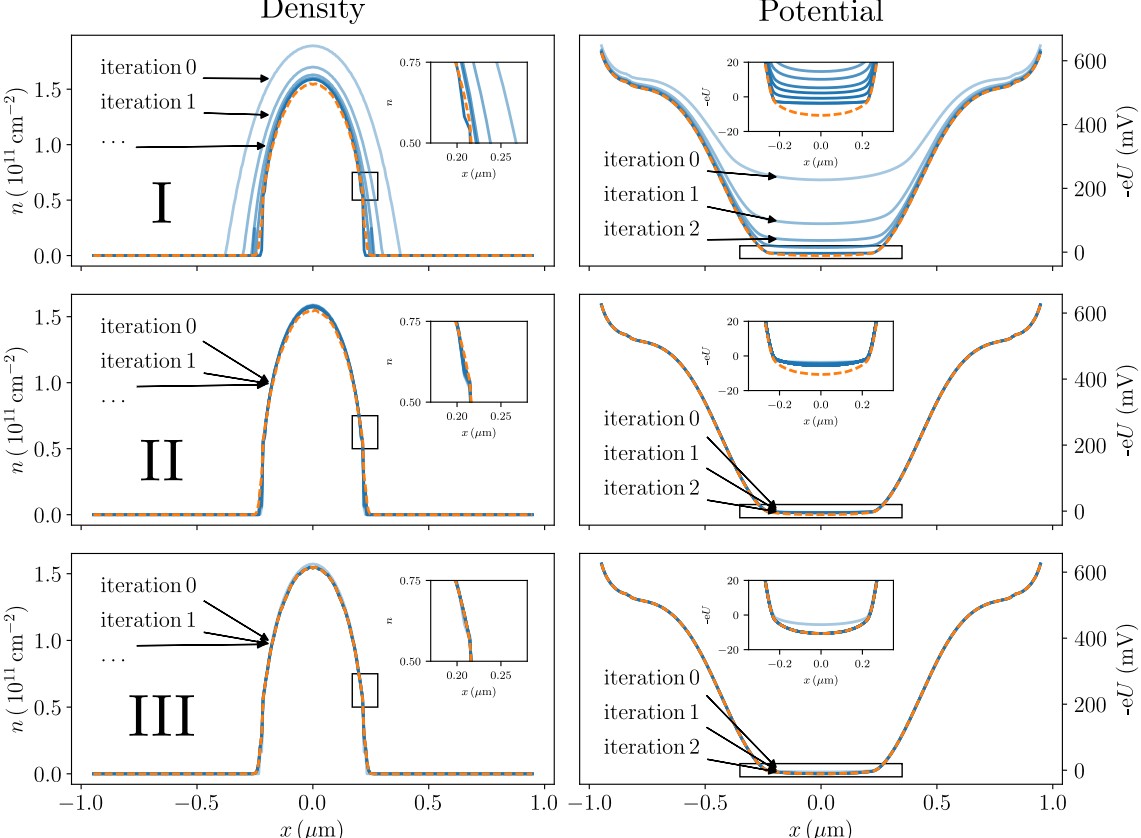

Figure 6: Relaxation of PAA and QAA for the geometry of Fig. 1(b). Left panels: density versus position. Right panels: electric potential versus position. Top panels: iterating over step **I** (excluding the sites with zero density). Middle panel: iterating over step **II** (relaxing the Poisson adiabatic approximation). Bottom panels: iterating over step **III** (relaxing the quantum adiabatic approximation) with several steps **II** (not shown) performed after each step **III**. Blue lines: various iterations. Thick orange dashed line: final converged result (FSC). Insets: zooms of the main curves.

the electronic gas where screening occurs. However, in regions where the electronic gas has been depleted ($n_i = 0$) there is no screening, hence the electric potential changes abruptly and the PAA fails. This problem is readily solved however: since we already know the density on these sites (it is zero), we do not need to solve a local adiabatic problem there. The first relaxation step, i.e. Step **I**, thus aims to detect such regions and remove them from the list of sites where the local self-consistent problem is solved. More precisely, we define the set $\mathcal{Q}' \subset \mathcal{Q}$ of sites where the density is non-zero and restrict the adiabatic self-consistent problem to $\mathcal{Q}'$. This has a strong influence on the electrostatics since the local capacitances $C_i$ strongly depend on the partitioning of $\mathcal{Q}$ into $\mathcal{Q}'$ and $\mathcal{Q} \setminus \mathcal{Q}'$. Indeed, since the PAA approximation is no longer performed on the sites belonging to $\mathcal{Q} \setminus \mathcal{Q}'$, their electrostatics is treated exactly. Hence the solution of the new adiabatic self-consistent problem on $\mathcal{Q}'$ results in an updated solution. Note that in the new solution some sites $i$ may become depleted and hence the set $\mathcal{Q}'$ must be updated again. This is achieved by performing step **I** once again. The procedure is repeated a few times until the set $\mathcal{Q}'$ no longer evolves. We emphasize that only a finite number of iterations of step **I** are needed to obtain the final set $\mathcal{Q}'$ (typically less than five). These iterations are computationally non-demanding since the same LDOS is used for all of them. As we shall see, the electronic density obtained after their completion is

almost indistinguishable from the FSC solution of the exact problem.

**II.** The purpose of the second kind of step is to relax PAA on the remaining sites $i \in \mathcal{Q}'$. This is achieved by solving the exact Poisson problem: given a density $n$ (such as the one obtained at the last step **I** iteration), one calculates the exact potential $U$, solution of the Poisson equation $n = CU$. The density $n$ is the new source term $n_i[U_i]$ in Eq.(16) and the potential $U$ serves as the new reference potential. Once Eq.(16) has been updated, we can solve the corresponding adiabatic self-consistent problem. Step **II** can be repeated until convergence. Note that, in practice, the Poisson equation and local capacitance are obtained by solving the mixed Poisson problem as explained below in Sec. 6. Typically very precise convergence is obtained within one or two step **II** iterations.

**III.** A third kind of step relaxes the QAA on the sites $i \in \mathcal{Q}'$. This is achieved by re-solving the quantum problem to update the LDOS. The new LDOS is integrated to update Eq. (12). Once Eq. (12) has been updated, we can solve the corresponding adiabatic self-consistent problem. Typically, we find that performing a single step **III** is sufficient. Calculating the ILDOS is the computational bottleneck of the calculation.

We emphasize that the relaxation steps **I**, **II** and **III** can, in principle, be performed in any order or even simultaneously. The most important one is step **I**, which is also the cheapest computationally. Hence, it should be performed first until convergence. Step **III** is far more computationally demanding than **I** or **II** since it implies solving the quantum problem. Hence, to optimize the number of step **III** iterations, it is preferable to first achieve convergence of step **II**. After each step **III** iteration, several step **II** iterations should be performed. Also, after each step **III** iteration, we reset step **I**, i.e. set $\mathcal{Q}' \equiv \mathcal{Q}$ and perform the step **I** relaxation until convergence. This is usually not needed but guarantees that the algorithm does not get trapped in a wrong $\mathcal{Q}'$ partition. After this sequence of relaxation steps, the final (supposedly exact) result is the FSC (Full Self-Consistent) solution to the quantum-electrostatic problem and is free from any initial approximations. We note that we have used plain iteration steps **II** and **III**. The relaxation could possibly be further accelerated by using mixing schemes such as DIIS, Anderson or Broyden algorithms.

Fig. 6 shows an example of performing several iterations of step **I** (upper panels), **II** (central panels) and **III** (lower panels) for the geometry of Fig. 1(b). The left panels show the density while the right panels show the potential. After each step **III**, a few steps **II** are performed. In most panels the curves for various iterations are almost superposed. The insets show zooms of the main curves which are also mostly superposed. The final converged FSC result is shown by a dashed orange curve. For the initial LDOS, we used the bulk (constant) DOS that is known analytically. In this case, it does not depend on energy. As anticipated, we observe that the initial solution of the adiabatic self-consistent problem is of bad quality, an indication that the PAA is a bad approximation in the depleted regions where the electric potential varies abruptly. However, after the vanishing density sites have been removed from the set of active sites $\mathcal{Q}'$ (after convergence of the steps **I**, cf. upper panels), we find that the density is almost indistinguishable from the final converged FSC result. We still observe a small (a couple of mV) discrepancy in the electric potential (see the zoom of the upper right panel). While this discrepancy is small on the global scale of Fig. 6, it is still important for quantitative transport calculations (cf. Sec. 10). The central panels illustrate the evolution of the solution upon performing several steps **II**. One observes that by relaxing the PAA, the results only change very slightly. This confirms that the PAA is an extremely good approximation inside the 2DEG. Since the bulk DOS was initially used, the results obtained after the steps **II** correspond to a self-consistent Thomas-Fermi calculation. In the last (lower) panel we perform the step **III** where the ILDOS is recalculated to relax the QAA. We find that one unique step is sufficient to obtain a fully converged result.

# 6 A mixed Neumann-Dirichlet Poisson solver

In usual electrostatic problems, one calculates some elements of the Green's function $G$. Indeed, in the standard Poisson problem one uses the density $n_\mu$ as an input and calculates the potential $U = Gn$ as an output. The Poisson problems that are repeatedly solved in our algorithm, however, involve elements of the *capacitance* matrix $C$. In this section, we explain how to formulate and solve a generalized Poisson problem that provides direct access to the relevant elements of the capacitance matrix $C$.

## 6.1 Problem formulation

We begin by sorting the sites of the set $\mathcal{P} = \mathcal{D} \cup \mathcal{N}$ into two categories that we call "Dirichlet" sites (set $\mathcal{D}$) and "Neumann sites" (set $\mathcal{N}$) in reference to the corresponding boundary conditions. The set $\mathcal{N}$ contains the sites where the density is an input and we want to calculate the potential. Therefore, $\mathcal{N}$ contains all the sites inside the dielectric (zero density) as well as the sites with dopants (known density). The depleted sites of the quantum problem $\mathcal{Q} \setminus \mathcal{Q}'$ are also elements of $\mathcal{N}$. The set $\mathcal{D}$ contains the sites where the potential is an input and we want to calculate the density. Hence, for the calculation of the local capacitances, $\mathcal{D}$ contains all the sites where the adiabatic self-consistent problem is defined $\mathcal{Q}' \subset \mathcal{D}$. Moreover, the sites that correspond to electrostic gates (standard Dirichlet boundary conditions) also belong to $\mathcal{D}$.

Writing Eq. (5) in a block form for the Dirichlet (D) and Neumann (N) blocks, it reads

$$\begin{bmatrix} \Delta_{NN} & \Delta_{ND} \\ \Delta_{DN} & \Delta_{DD} \end{bmatrix} \cdot \begin{bmatrix} U_N \\ U_D \end{bmatrix} = \begin{bmatrix} n_N \\ n_D \end{bmatrix}. \tag{19}$$

In the above equation $n_N$ and $U_D$ are the known inputs of the problem while $n_D$ and $U_N$ are yet to be determined. After reshuffling the above equation, we arrive at the "mixed Neumann-Dirichlet Poisson problem",

$$\begin{bmatrix} \Delta_{NN} & 0 \\ \Delta_{DN} & -\mathbb{1} \end{bmatrix} \cdot \begin{bmatrix} U_N \\ n_D \end{bmatrix} = \begin{bmatrix} \mathbb{1} & -\Delta_{ND} \\ 0 & -\Delta_{DD} \end{bmatrix} \cdot \begin{bmatrix} n_N \\ U_D \end{bmatrix}. \tag{20}$$

Solving this problem amounts to solving a set of linear equations with the right-hand side as a source term. This is readily achieved with sparse solvers such as the MUMPS package [31]. Two different quantities must be calculated with the mixed Poisson solver, respectively the source term $n_i[U]$ and the local capacitance $C_i$.

To calculate $n_i[U]$, one sets $n_N$ and $U_D$ to their known values. The density $n_N$ is the result of the local self-consistent calculations. $U_D$ is equal to the input gate potential at the electrostatic gates.

To calculate the vector $C_i$ for $i \in \mathcal{Q}'$, one sets $n_N = 0$ and $U_D = 0$ except for sites $i \in \mathcal{Q}'$ where $U_i = 1$. The output vector $n_D$ (projected on $\mathcal{Q}'$) contains the needed elements $C_i = (n_D)_i$.

## 6.2 Finite volume discretization

In order to obtain the $\Delta_{\mu\nu}$ matrix from the continuum problem, a discretization scheme of some sort must be used. Many approaches could be employed, including finite-difference and finite-elements methods. Here we use a finite-volume approach that has the advantage of solving a problem which is physically meaningful for any finite value of the discretization length $a$. In particular, this method has the advantage of respecting charge conservation inside the Neumann sites independently of the discretization length $a$. Since the quantum-electrostatic

problem is extremely sensitive to any variation of the charge, strict charge conservation is very important to ensure fast convergence with respect to $a$.

One starts by meshing the simulation box to obtain the $\mathcal{P}$ sites. One includes all the sites $\mathcal{Q}$ of the quantum system (this is important to avoid any interpolation difficulty between the quantum and Poisson problem). Then one adds a regular grid around the quantum sites. This grid matches the lattice of the quantum system to avoid introducing artificial noise due to lattice mismatch. Another grid with a larger value of $a$ can be used far away from the quantum system.

In a second step one constructs the Voronoi cells associated with our mesh using the Qhull algorithm [32]. An example of the final discretized geometry with the Voronoi cells is shown in Fig. 7 for the system of Fig. 1(b). For clarity, Fig. 7 shows very few cells. In actual calculations we use meshes with typically $10^4$ sites in 2D and $10^6$ sites in 3D.

To calculate the $\Delta_{\mu\nu}$ matrix, we apply the Gauss theorem to each volume cell. One obtains, without approximation,

$$n_\mu = \sum_\nu \Phi_{\mu\nu}, \tag{21}$$

with $n_\mu$ the total charge inside the cell.

$$\Phi_{\mu\nu} = \int_{S_{\mu\nu}} \epsilon(\vec{r})\, \vec{E}(\vec{r}) \cdot \vec{n}\, \mathrm{d}S \tag{22}$$

is the flux of the electric field $\vec{E}$ through the planar surface $S_{\mu\nu}$ that connects cell $\mu$ with cell $\nu$ ($\vec{n}$ is the unit vector point perpendicular to this surface). In the electrostatic limit, the electric field is irrotational which reads

$$\oint_{\vec{r}_\mu}^{\vec{r}_\nu} \mathrm{d}\vec{r} \cdot \vec{E} = U_\mu - U_\nu. \tag{23}$$

To close our system of equations, we suppose that the electric field varies smoothly on the scales of the Voronoi cell. Up to $O(a^3)$ corrections one obtains

$$\Phi_{\mu\nu} = \frac{\epsilon_{\mu\nu} S_{\mu\nu}}{d_{\mu\nu}} \left(U_\mu - U_\nu\right), \tag{24}$$

where $d_{\mu\nu}$ is the distance between the center of the two cells and $\epsilon_{\mu\nu} = 2\epsilon_\mu \epsilon_\nu/(\epsilon_\mu + \epsilon_\nu)$ is an averaged dielectric constant obtained from the conservation of the flux through the surface. Together, Eq. (21) and Eq. (24) define the $\Delta_{\mu\nu}$ matrix.

# 7 Calculation of the integrated local density of states

Solving the local quantum problem obtained with the Quantum adiabatic approximation implies calculating the ILDOS as a function of the chemical potential for every site of the quantum system $\mathcal{Q}$. The numerical integration of the LDOS can be difficult in some situations: one example is shown in Fig. 3. There, the LDOS has singularities at zero fields and Dirac functions in presence of magnetic field. A direct calculation of the integral of Dirac functions using quadrature rules is bound to failure. In this section we explain how to circumvent this problem using quadrature methods over momentum instead of energy. We note that a popular approach to calculate the density uses complex contour integration with, for instance, the so-called Ozaki contour [33]. Although this method works very well at equilibrium (but not out-of-equilibrium), it is unsuitable for our purpose as it provides the density for a single

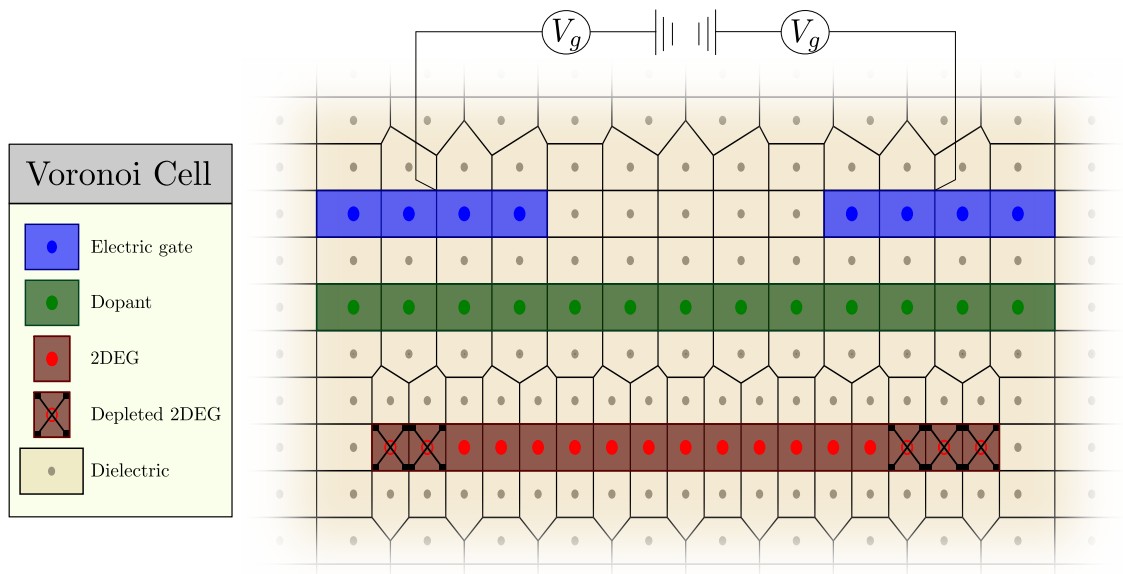

Figure 7: Sketch of the discretized Poisson problem for the geometry of Fig. 1(b). The 2DEG and depleted 2DEG voronoi cells belong to the sites in $\mathcal{Q}'$ and $\mathcal{Q} \setminus \mathcal{Q}'$ respectively. The dopant and dielectric voronoi cells correspond to Neumann sites, i.e. belong to $\mathcal{N} \subset \mathcal{P}$. The cells forming the electric gate belong to $\mathcal{D} \subset \mathcal{P}$.

value of the chemical potential. Indeed, we require the full function ILDOS versus chemical potential to solve the adiabatic self-consistent problems.

For simplicity, we restrict ourselves to calculations at zero temperature (the method is readily extended to arbitrary temperatures). The ILDOS on site $i \in \mathcal{Q}$ is defined as

$$n_i[\mu] = \int^{\mu} \mathrm{d}E \ \rho_i(E), \tag{25}$$

where the lower bound of the integral is the beginning of the spectrum. The LDOS $\rho_i(E)$ is itself defined in terms of the wave functions of the system with momentum $k$ as,

$$\rho_i(E) = \int_{-\pi}^{\pi} \frac{\mathrm{d}k}{2\pi} \sum_{\alpha} |\psi_{\alpha k}(i)|^2 \delta[E - E_{\alpha}(k)], \tag{26}$$

where $E_{\alpha}(k)$ is the dispersion relation of the corresponding band. The above expression is valid for translational invariant systems such as the geometry of Fig. 1(b). For more general geometries, such as Fig. 1(c), the momentum $k$ is to be understood as the momentum in the semi-infinite electrodes. To calculate the ILDOS, we insert Eq. (26) into Eq. (25) and invert the order of the integrals. The integral over energy can be performed exactly and we arrive at

$$n_i[\mu] = \int_{-\pi}^{\pi} \frac{\mathrm{d}k}{2\pi} \sum_{\alpha} |\psi_{\alpha k}(i)|^2 \theta[\mu - E_{\alpha}(k)], \tag{27}$$

where $\theta(x)$ is the Heaviside function. Eq. (27) can now be evaluated by standard quadrature techniques that sample the $k$ points. One can readily understand why this change of variable $E = E_{\alpha}(k)$ is particularly advantageous in the case of the quantum Hall effect. There, the dispersion relation $E_{\alpha}(k)$ is extremely flat due to the presence of the dispersion-less Landau levels. By sampling in $E$-space, one is almost certain not to sample correctly these Landau levels. By sampling in $k$-space, however, the points get automatically positioned where they

are needed. Furthermore, the integral for many values of $\mu$ is performed simultaneously at no additional computational cost.

While the above scheme provides the exact ILDOS, one could also consider using approximate (but computationally less intensive) forms of the ILDOS. An obvious choice it to use the bulk DOS as the LDOS. This leads to the Thomas-Fermi approximation. One could also use the adiabatic approximation such as in [4] where the 3D LDOS is replaced by the solution of 2D problems that depend on the third dimension. Iterative methods such as the Kernel Polynomial Method (KPM) are also natural approaches for obtaining the ILDOS [34].

# 8 Summary of the Algorithm

Let us summarise the different phases of our method. Fig. 8 shows the corresponding flowchart. First, the self-consistent adiabatic problem must be initialized. To initialize the ILDOS one can solve the quantum problem with a vanishing electric potential $U = 0$. Calculating the ILDOS, however, is the most computationally expensive step in the algorithm. Therefore it is often more efficient to intialize the ILDOS with the bulk value for the material. This corresponds to the Thomas-Fermi approximation. Using such an initial ILDOS provides an accurate electronic density and allows one to reduce the number of quantum calculations (step **III**) by one. The algorithm is usually not sensitive to how the Poisson problem is initialized. One calculates the initial source term $n_i(U)$ by supposing e.g. an absence of screening (no charge in the quantum part $\mathcal{Q}$). The local capacitances $C_i$ are calculated assuming that all quantum sites are active ($\mathcal{Q}' = \mathcal{Q}$). Once the adiabatic self-consistent problem has been constructed, it is solved on all active sites by finding intersections between 1D functions.

In a second phase, the ILDOS (step **III**), the source density $n_i(U)$ (step **II**) and the list of active sites (step **I**) must be updated until convergence to the FSC solution. We have found that the order in which the steps **I**, **II** and **III** are performed is not critical. Fig. 8 shows the flowchart that we have used in this article. It aims at minimizing the number of computationally intensive steps **III**. Following the flowchart one starts by repeating step **I** until the $\mathcal{Q} \setminus \mathcal{Q}'$ decomposition has converged. Then one iterates over step **II** until the local poisson problem has converged. Finally, one iterates over step **III** until the integrated local density of states (ILDOS) has converged. After each iteration of step **I**, **II** or **III** the adiabatic self-consistent problem is updated, solved and its convergence verified. Once the $\mathcal{Q} \setminus \mathcal{Q}'$ decomposition, the local poisson problem and the ILDOS have converged to a desired accuracy, the result of the adiabatic self-consistent problem can be used to calculate observables. The latter can be, for example, the local current density or the conductance, such as calculated in Sec. 9 and Sec. 10

# 9 Application to the quantum Hall effect

We are now ready to apply our algorithm to situations where the density of states varies abruptly, i.e. when the quantum-electrostatic problem is highly non-linear. A rather extreme situation is the quantum Hall effect where the ILDOS has the staircase shape shown in Fig. 3. In this section, we consider the geometry of Fig. 1(b) in presence of a perpendicular magnetic field. The physics contained here has been discussed in a separate paper [35]. The results shown below aim at illustrating the algorithm as well as providing additional data that were not shown in [35].

Fig. 9 shows the electronic density (top), electric potential (middle) and band structure (bottom) for three values of the magnetic field (left, middle and right). The blue curves correspond to the Thomas-Fermi approximation, i.e. to solving the self-consistent problem with

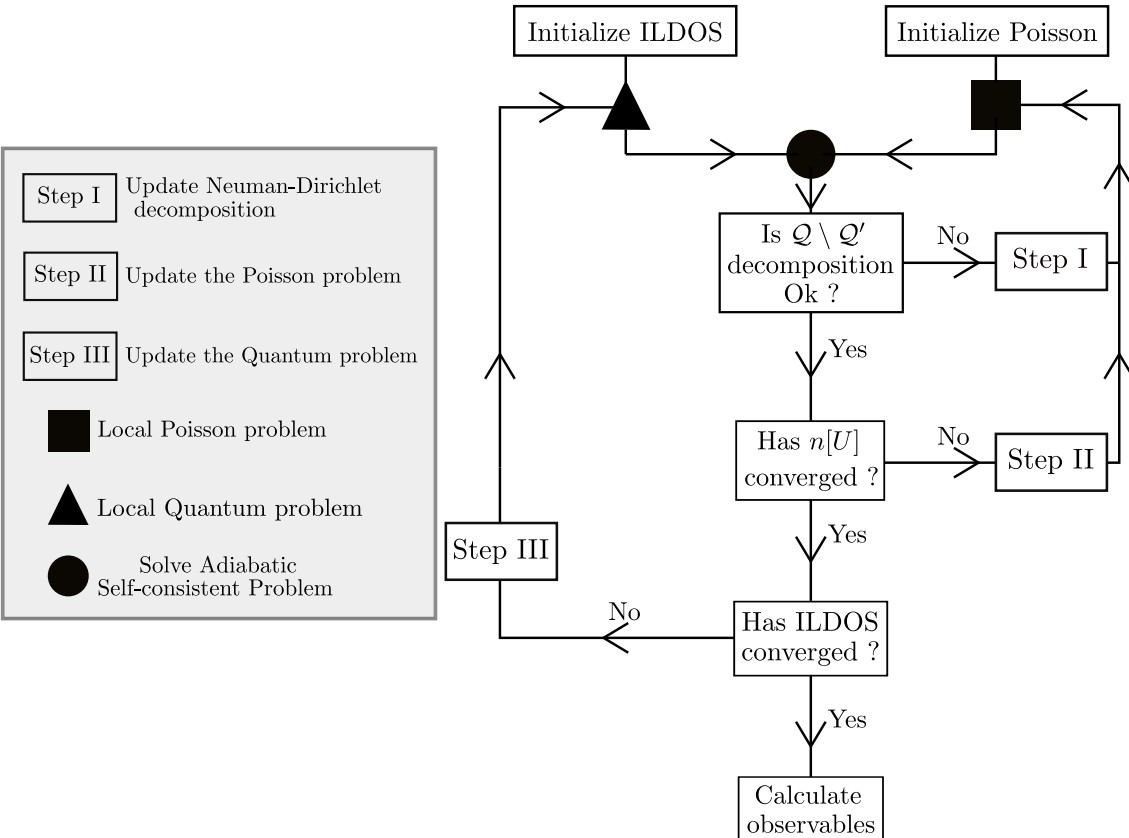

Figure 8: Flowchart for the relaxation steps **I**, **II** and **III**.

the ILDOS of an infinite bulk system (perfect staircase of Landau levels). In the Thomas-Fermi approximation, we recover the Shklovskii-Chklovskii-Glazmann picture of compressible and incompressible stripes [36,37]. The compressible stripes are regions of constant potential and varying density while the incompressible stripes are zones of constant density and varying potential. In the incompressible stripes, there are no accessible states at the Fermi energy. We further observe that the full self-consistent solution (orange lines) is significantly different from the Thomas-Fermi approximation. In particular, the steps in density are no longer present and the ones in the potential only appear at high enough field.

The positions $x_\nu$ of the center of the incompressible stripes can be estimated from the electronic density calculated at zero field $n(x, B = 0)$ since, with very good approximation, $n(x_\nu, B = 0) = \nu e B/h$ with $\nu = 1, 2, \ldots$. The width $\delta x_\nu$ of these plateaus can also be estimated using a simple energetic argument. The creation of the incompressible stripe involves the creation of the small electric dipole of charge $\delta q_\nu$ with respect to the $B = 0$ density. On one hand, we have $\delta q_\nu \approx e \partial_x n(x_\nu, B = 0) \delta x_\nu$. On the other hand, electrostatics imposes $\delta q_\nu \approx c(\epsilon/\delta x_\nu)(\hbar \omega_c/e)$. Here $c(\epsilon/\delta x_\nu)$ is the effective capacitance of the problem and $(\hbar \omega_c/e)$ is the kinetic energy gained by creating the stripe which compensates the corresponding electric energy ($\omega_c = eB/m^\star$ is the cyclotron frequency). We arrive at [36]

$$\delta x_\nu \approx \sqrt{\frac{c \epsilon \hbar B}{e m^* \partial_x n(x_\nu, B = 0)}}, \tag{28}$$

with the constant given by $c \approx 5.1$ for our particular geometry. To verify the above expression, Fig. 10 shows the gradient of the density $\partial_x n$ (top panels) and of the electric potential $\partial_x U$ (bottom panels) as a function of position $x$ and magnetic field $B$. The gradients vanish for the incompressible and compressible stripes, respectively. Left and right panels correspond

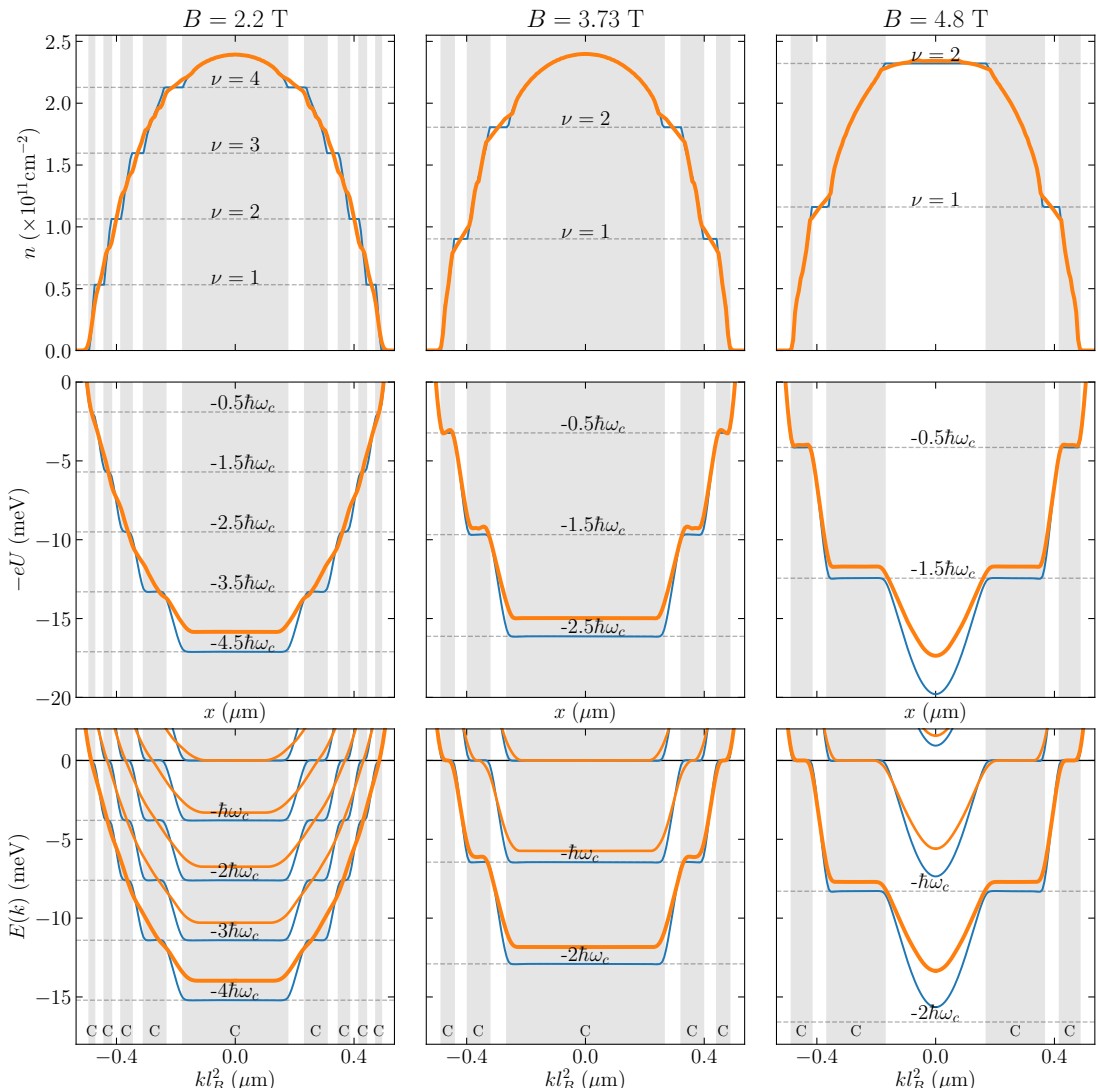

Figure 9: Density (top), potential (middle) and band structure (bottom) for different magnetic fields (from left to right $B = 2.2, 3.73, 4.8$T) for the geometry of Fig. 1(b). Thomas-Fermi approximation is shown in blue lines while the Full self-consistent (FSC) result is shown in thicker orange lines. The horizontal lines indicate the integer filling factors $\nu$ ($n = \nu eB/h$) and the multiple of the cyclotron frequency $\omega_c = eB/m^\star$. Gray and white regions indicate the compressible and incompressible stripes, respectively.

respectively to Thomas-Fermi and FSC which are difficult to distinguish at this scale. The different types of stripes are easy to identify. The dashed line corresponds to the width predicted with Eq. (28) which match the numerics quantitatively.

We now proceed to the calculation of the conductance of a ballistic conductor. Assuming all channels are perfectly transmitted (no reflection), the Landauer formula takes the particularly simple form

$$g = -2e^2 \sum_\alpha \int \frac{\mathrm{d}k}{2\pi} \, \theta(v_{\alpha k}) v_{\alpha k} \frac{\partial f}{\partial E} [E_\alpha(k)], \tag{29}$$

where $v_{\alpha k} = \partial_k E_\alpha(k)/\hbar$ is the velocity of the mode $\alpha$ at the Fermi energy and the Heavyside

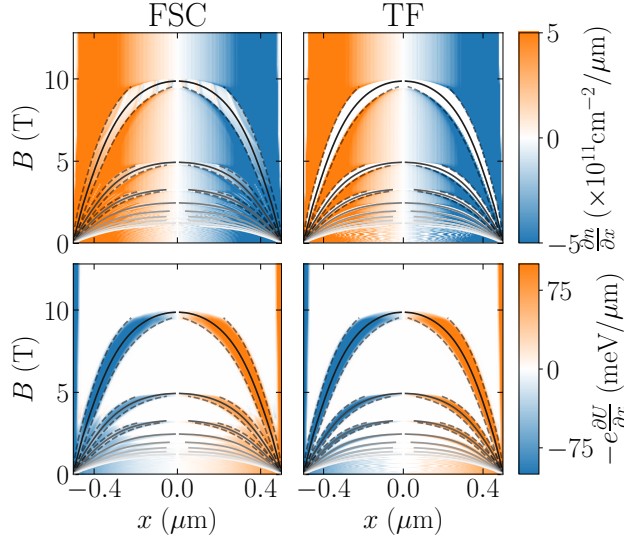

Figure 10: Gradient of the density (top) and potential (bottom) as a function of the magnetic field $B$ at $V_g = -0.75$V for the geometry of Fig. 1(b). Left panel: FSC, right panels: Thomas-Fermi. The black lines correspond to the theoretical estimate for $x_\nu$ while the dashed lines correspond to $x_\nu \pm \delta x_\nu / 2$ for the first three Landau levels $\nu = 1, 2, 3$.

function selects channels with positive velocity. The conductance obtained as a function of the gate voltage for two temperatures and magnetic fields are shown in Fig. 11. These calculations show the crossover between channel quantization at low field and the quantum Hall effect at high field. It is interesting to note that the presence of a degenerate band at the Fermi level in the quantum Hall regime leads to a non-quantized conductance even though the system is perfectly ballistic. The dashed orange line shows the corresponding estimate $g = n(x = 0, B = 0)e/B$ which fits fairly well the conductance outside of the plateaus.

We proceed with the calculation of the local density of current $J(x)$ which is given by

$$J(i) = -2e^2 \sum_\alpha \int \frac{\mathrm{d}k}{2\pi} |\psi_{\alpha k}(i)|^2 \theta(v_{\alpha k}) v_{\alpha k} \frac{\partial f}{\partial E}[E_\alpha(k)]. \tag{30}$$

The dependance of $J(x)$ as a function of position and magnetic field is shown in the colormap in Fig. 12. Note that we only discuss the out-of-equilibrium current. In the quantum Hall regime there is also an equilibrium current flowing in the incompressible stripes. Here, it has been subtracted. Fig. 12 provides the answer to a small paradox: in incompressible stripes there are no available states at the Fermi level, and thus no out-of-equilibrium current can flow in these zones. Therefore, the current can only flow in the compressible regions. However, in the latter the dispersion relation is flat, hence the states have vanishing velocity, and thus vanishing currents. Therefore it would naively seem that no out-of-equilibrium current can flow in the system. This paradox is only present in the Thomas-Fermi picture. Indeed, the FSC calculations clarify the question of where the current flows. Unsurprisingly, we find that the current density lies mostly at the boundary between compressible and incompressible stripes.

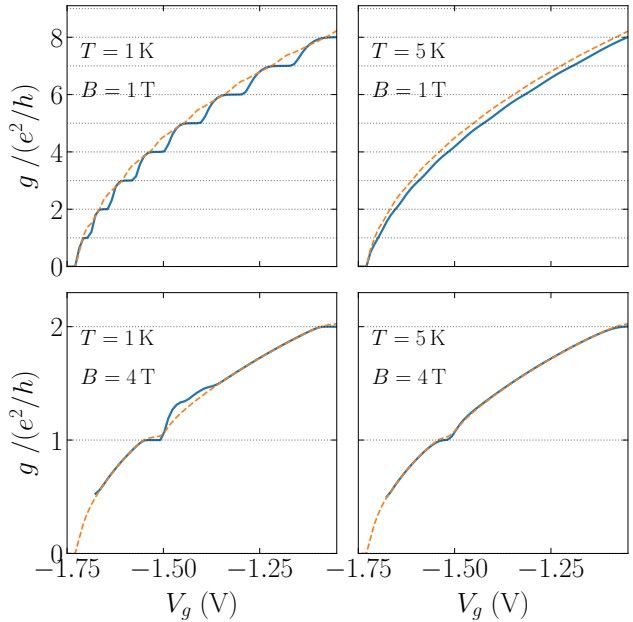

Figure 11: Conductance $g$ in units of $e^2/h$ as a function of the gate voltage $V_g$ for two different magnetic fields $B = 1$ T and $B = 4$ T (up and bottom panels) and 2 different temperatures $T = 1$ K and $T = 5$ K (left and right panels). Solid blue line: FSC calculation. Dashed orange line: $g = n(x = 0, B = 0)e/B$. Dotted horizontal thin lines correspond to quantized values of the conductance.

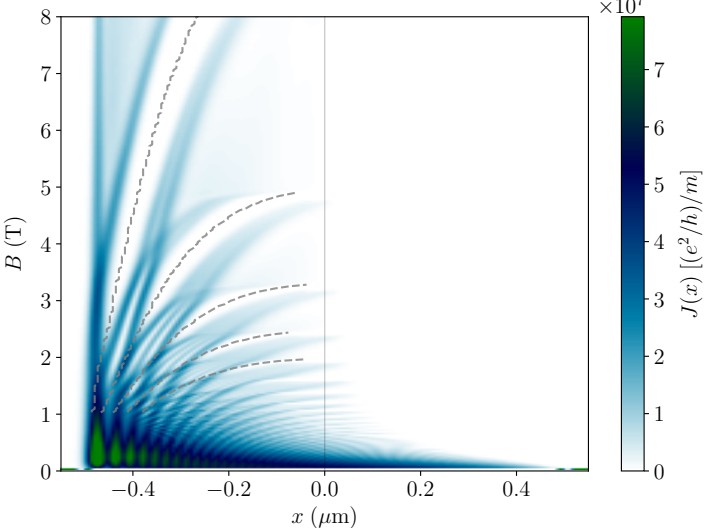

Figure 12: Map of the current density $J(x)$ as a function of the position $x$ and the magnetic field $B$. Dotted lines indicate the centers of the incompressible stripes.

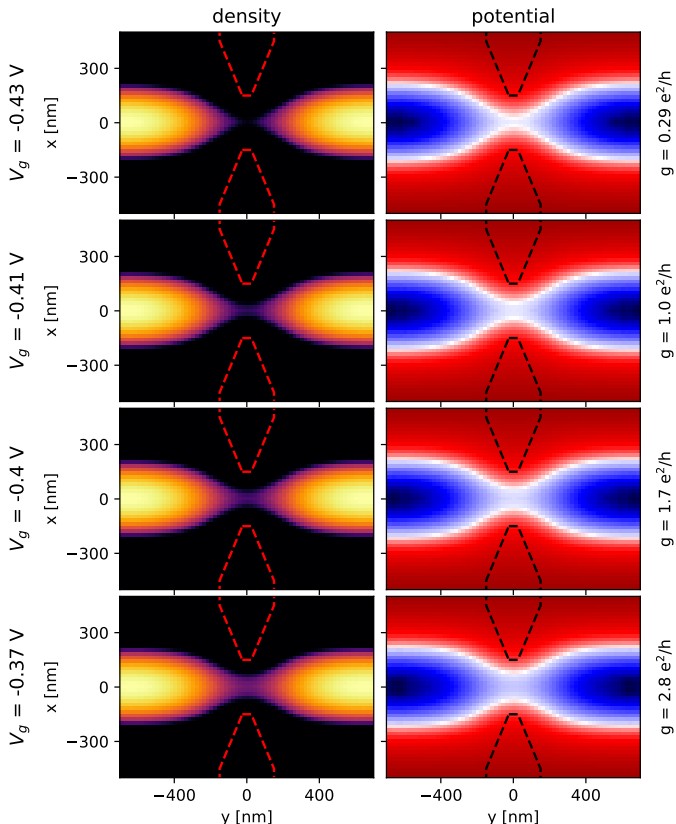

Figure 13: 2D maps of the density (left) and electric potential (right) for four different gate voltages $V_g = -0.43, -0.41, -0.4$ and $-0.37$V (top to bottom) for a QPC. Left: density of the 2DEG (black corresponds to zero density). Right: electric potential (blue corresponds to negative potential where the 2DEG lies, white is zero and red corresponds to positive potential where the 2DEG is depleted). An additional side gate $V_s = -0.8$ depletes the gas far away from the QPC. The dashed lines indicate the gates of the QPC. The scale of variation of the potential is shown in Fig. 14

## 10  Applications to a quantum point contact

We now turn to a second application, the study of the quantum point contact (QPC) geometry of Fig. 1(c). QPCs are important historically as the first device where conductance quantization was observed [38,39]. They can be considered as the electronic equivalent of the optical beam splitter and as such play a central role in electronic quantum optics [40].

Fig. 13 shows colormaps of the density and electric potential around the QPC for different values of the confining gate potential $V_g$ applied to the QPC. These FSC results correspond to a 2D quantum problem with around $10^4$ active quantum sites ($\mathcal{Q}$ sites) embedded in a 3D Poisson problem with around $10^6$ electrostatic sites ($\mathcal{P}$ sites). Fig. 14 shows cuts of the colormap at various positions. The electron gas is present in regions where the electric potential is negative. Typical values of the potential in these regions is of a few mV. Convergence with an accuracy better than $10\mu$V is needed around the QPC to obtain reliable results for transport calculations.

We end this section with the calculation of the conductance versus gate voltage, the actual observable measured in most experiments. The results are shown in Fig. 15 for various itera-

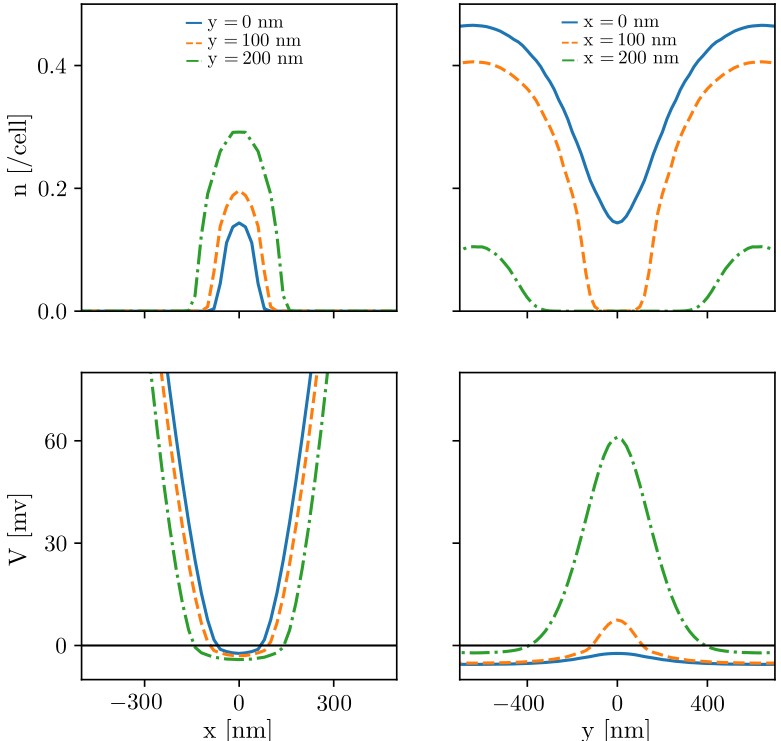

Figure 14: Cut of Fig. 13 at constant $x$ (left) and $y$ (right) for the density (upper panels) and potential (lower panels). The cut correspond to $x, y = 0$ (blue), 100 (dashed orange) and 200 nm (dot dashed green). The QPC gate voltage is set at $V_g = -0.37$ V

tions of step **III**. Each iteration corresponds to a new calculation of the ILDOS. Iteration 0 is the Thomas-Fermi approximation. It provides an accurate density but the $g(V_g)$ curve is not quantitative (offset of the pinch voltage of 0.2V and wrong size of the conductance plateaus). The results are fully converged after a single iteration of the ILDOS. These calculations, which map the input experimental parameters to the experimental observables, are directly comparable to experiments [41].

## 11  Conclusion

We have developed a new algorithm that is able to solve the quantum-electrostatic problem even in highly non-linear situations. Perhaps more importantly, we have observed that the algorithm converges extremely rapidly without requiring any parameter tuning. This is true even at zero temperature and/or under high magnetic field. This opens the possibility for direct and detailed comparisons between experiments and simulations, a prerequisite for using simulations at the design stage of quantum devices.

## Acknowledgements

We thank Y-M Niquet, M. Wimmer and A. Akhmerov for interesting discussions. This project is funded by the US Office of Naval Research, the French-USA ANR PIRE, the French-Japon

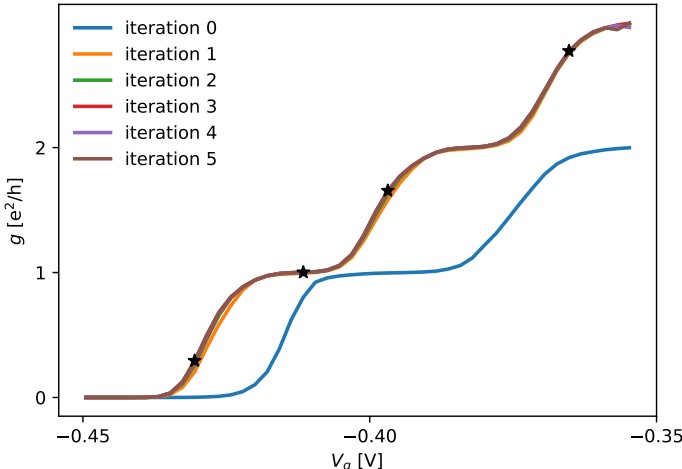

Figure 15: Conductance of a QPC in units of $e^2/h$ as a function of the gate voltage $V_g$ for different quantum iterations (QAA). The zeroth iteration (blue line) corresponds to Thomas Fermi calculations. The black star indicates the chosen voltages and conductances for Fig 13.

ANR QCONTROL and the E.U. flagship graphene (ANR GRANSPORT).

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
