# Peer review of "The self-consistent quantum-electrostatic problem in strongly non-linear regime"

_SciPost Physics, doi:SciPost Phys. 7, 031 (2019)_

## Round 1 · Referee Report · Anonymous (Referee 1) · 2019-5-8

Strengths

  1. Clearly written
  2. Well presented and documented
  3. Detailed discussion of a difficult problem
  4. Valuable information for researchers discussed in one place
  5. New stable approach to a known problem

Weaknesses

No obvious weakness

Report

The authors propose an effective method to solve "the
the self-consistent quantum-electrostatic problem" at
zero temperature. Commonly, this has been avoided by
performing the calculations at a finite, but low
temperature. Indeed, the suggested methodology is
well explained, documented, and supported by examples.
All this together makes the manuscript very valuable
to many researchers that have to address this problem
in some form in their modeling of physical phenomena.

Requested changes

For the authors to consider:

1.
The manuscript is well written, but I am note quite
sure if in the sentence:

"The technique is intrinsically convergent including
in highly non-linear regimes."

the word "including" is the best choice.

2.
The authors do investigate the "self-consistent
quantum-electrostatic problem", which they also
refer to as the "Poisson-Schrödinger problem".
In view of a difference in use of the terms within
the physics and the mathematical communities I would
like the authors briefly to relate this problem to
the Hartree Approximation (HA) which in the physics
and the quantum chemistry communities is also the
mentioned simultaneous solution of the Schrödinger
and the Poisson equations with the condition that
the wavefunctions are to be orthogonal.
In the mathematics community the problem is often
confronted without this last condition.
The different stand point can be referred back to
the need of physicists to relate the HA to higher
order ones, the Hartree-Fock Approximation (HFA), or
higher order Green functions schemes. This issue
is briefly mentioned in the Introduction of:
The European Physical Journal B 84, 699 (2011).

  • validity: top
  • significance: high
  • originality: high
  • clarity: top
  • formatting: excellent
  • grammar: excellent

Author:  Xavier Waintal  on 2019-06-28  [id 551]

(in reply to Report 1 on 2019-05-08)
Category:
remark

We thank our referees for the report. In response to his/her comments,

  • we now mention the self-consistent Hartree approximation as an equivalent name in the introduction.

  • We have replaced the word "including" by "even" in the abstract.

---

## Round 1 · Referee Report · Anonymous (Referee 2) · 2019-6-5

Report

In this manuscript, the authors present the computational scheme to perform Schrodinger-Poisson simulations of nanoscale quantum systems. The main result of the manuscript is eliminating the convergence issues of the Schrodinger-Poisson problem and reducing the necessity to perform computationally intensive steps of quantum simulation to only a few iterations.

The framework consists of several steps: - Inverting Poisson equation as a capacitance equation for the density as a function of potential, thus rewriting the original problem to that of finding the intersection between solutions of inverted Poisson and Schrodinger equations. The authors show that the capacitance is a better behaved numerical quantity than its inverse. - Developing a solver for inverted Poisson equation to obtain the capacitance and density for the given potential. - Performing adiabatic approximations for Poisson and Schrodinger equations, allowing for linearization of electronic density and reducing the intersection problem to a set of local problems that can be converged rapidly. - Performing a set of steps to get the adiabatic problem to converge to the full self-consistent solution. Additionally, the authors implement the finite volume method to discretize the Poisson equation, which allows them to enforce the charge conservation.

Schrodinger-Poisson is a nonlinear problem and thus has convergence issues. In fact, as also described in this paper, the speed of convergence decreases with the system size - the potential needs to be converged on all lattice sites. This paper eliminates this unwanted prefactor by mapping it to multiple 1d self-consistent equations. While the principal complexity of Schrodinger-Poisson method is in the cubic scaling of the calculation of density from the Schrodinger equation, this manuscript eliminates the additional overhead, significantly helping to achieve a truly realistic computational modeling. I am sure this paper is useful as a handbook for various implementations of the Schrodinger-Poisson method and thus should definitely be published in SciPost.

The paper is very clearly written, all the steps are explained in detail. Additionally to the typos noted by the other referee, several edits might improve the manuscript: 1. Since the main contribution of the manuscript is the algorithm, it would be worth spelling it out completely with all the steps in a subsection. 2. Please provide more details on the parametrization of the problem. Section 2 says "We assume … a 2DEG density $2.11 x 10^{11} cm^{-2]$". Since density is self-consistently evaluated and changes in the presence of gates, what is that density the authors refer to? Is it the density of dopants or is the dopant density fitted such as to have this density in 2DEG in the absence of external gates? I believe that both this and the companion paper would also benefit from writing down all the parameters and boundary conditions.

  • validity: top
  • significance: top
  • originality: top
  • clarity: top
  • formatting: perfect
  • grammar: perfect

Author:  Xavier Waintal  on 2019-06-28  [id 552]

(in reply to Report 2 on 2019-06-05)
Category:
remark

We thank our referee for the report. In response to his/her comment,

  • We have added a new section 8 "Summary of the Algorithm". This section contains a chart that explains how the different parts of the algorithm are articulated.

  • We have added a few more details on the parameters used in the simulations.

---

## Round 2 · Author Response

Dear Editor,

Please find the new version of our manuscript for resubmission.
The main modification we did was to add a section and a figure that
summarizes the flowchart of our manuscript.

Yours sincerely,
the authors

---

## Round 2 · List of Changes

- we now mention the self-consistent Hartree approximation as an equivalent name in the introduction. (referee 1 suggestion)

- We have replaced the word "including" by "even" in the abstract (referee 1 suggestion)

- We have added a new section 8 "Summary of the Algorithm" in response to our second report. This section contains a chart that explains how the different parts of the algorithm are articulated. (referee 2 suggestion)

- We have added a few more details on the parameters used in the simulations. (referee 2 suggestion)

---

## Editorial Decision

published